**A new multi-variable benchmark for Last Glacial Maximum climate simulations**

Sean F. Cleator[1], Sandy P. Harrison[2], Nancy K. Nichols[3], I. Colin Prentice[4] and Ian Roulstone[1]

1: Department of Mathematics, University of Surrey, Guildford GU2 7XH, UK
2: School of Archaeology, Geography and Environmental Science, University of Reading, Whiteknights, Reading, RG6 6AH, UK
3: Department of Mathematics & Statistics, University of Reading, Whiteknights, Reading RG6 6AX, UK
4: AXA Chair of Biosphere and Climate Impacts, Department of Life Sciences, Imperial College London, Silwood Park Campus, Buckhurst Road, Ascot SL5 7PY, UK

Journal: Climate of the Past

**Abstract**. We present a new global reconstruction of seasonal climates at the Last Glacial Maximum (LGM, 21,000 yr BP) made using 3-D variational data assimilation with pollen-based site reconstructions of six climate variables and the ensemble average of the PMIP3/CMIP5 simulations as a prior. We assume that the correlation matrix of the uncertainties of the prior both spatially and temporally is Gaussian, in order to produce a climate reconstruction that is smoothed both from month to month and from grid cell to grid cell. The pollen-based reconstructions include mean annual temperature (MAT), mean temperature of the coldest month (MTCO), mean temperature of the warmest month (MTWA), growing season warmth as measured by growing degree days above a baseline of 5°C (GDD$_5$), mean annual precipitation (MAP) and a moisture index (MI), which is the ratio of MAP to mean annual potential evapotranspiration. Different variables are reconstructed at different sites, but our approach both preserves seasonal relationships and allows a more complete set of seasonal climate variables to be derived at each location. We further account for the ecophysiological effects of low atmospheric carbon dioxide concentration on vegetation in making reconstructions of MAP and MI. This adjustment results in the reconstruction of wetter climates than might otherwise be inferred by the vegetation composition. Finally, by comparing the uncertainty contribution to the final reconstruction, we provide confidence intervals on these reconstructions and delimit geographical regions for which the palaeodata provide no information to constrain the climate reconstructions. The new reconstructions will provide a benchmark created using clear and defined mathematical procedures that can be used for evaluation of the PMIP4/CMIP6 entry-card LGM simulations and are available at DOI:10.17864/1947.229.

## 1 Introduction

Models that perform equally well for present-day climate nevertheless produce very different responses to anthropogenic forcing scenarios through the 21$^{st}$ century. Although internal variability contributes to these differences, the largest source of uncertainty in model projections in the first three to four decades of the 21$^{st}$ century stems from differences in the response of individual models to the same forcing (Kirtman et al., 2013). Thus, the evaluation of models based on modern observations is not a good guide to their future performance, largely because the observations used to assess model performance for present-day climate encompass too limited a range of climate variability to provide a robust test of the ability to simulate climate changes. Although past climate states do not provide analogues for the future, past climate changes provide a unique opportunity for out-of-sample evaluation of climate model performance (Harrison et al., 2015).

At the Last Glacial Maximum (LGM, conventionally defined for modelling purposes as 21 000 years ago), insolation was quite similar to the present, but global ice volume was at a maximum, eustatic sea level was close to a minimum, long-lived greenhouse gas concentrations were lower, and atmospheric aerosol loadings higher than today, and land surface characteristics (including vegetation distribution) were also substantially different from today. These changes gave rise to a climate radically different from that of today; indeed the magnitude of the change in radiative forcing between LGM and pre-industrial climate is comparable to high-emissions projections of climate change between now and the end of the 21$^{st}$ century (Braconnot et al., 2012). The LGM has been a focus for model evaluation in the Paleoclimate Modelling Intercomparison Project (PMIP) since its inception (Joussaume and Taylor, 1995; Braconnot et al., 2007; Braconnot et al., 2012). The LGM is one of the two "entry card" palaeoclimate simulations included in the current phase of the Coupled Model Intercomparison Project (CMIP6) (Kageyama et al., 2018). The evaluation of previous generations of palaeoclimate simulations has shown that the large-scale thermodynamic responses seen in 21$^{st}$ century and LGM climates, including enhanced land–sea temperature contrast, latitudinal amplification, and scaling of precipitation with temperature, are likely to be realistic (Izumi et al., 2013; Li et al., 2013; Lunt et al, 2013; Hill et al., 2014; Izumi et al., 2014; Harrison et al., 2015). However, evaluation against palaeodata shows that even when the sign of large-scale climate changes is correctly predicted, the patterns of change at a regional scale are often inaccurate and the magnitudes of change often underestimated (Brewer et al., 2007; Mauri et al., 2014; Perez Sanz et al., 2014; Bartlein et al., 2017). The current focus on understanding what causes mismatches between reconstructed and simulated climates is a primary motivation for developing benchmark data sets that represent regional climate changes comprehensively enough to allow a critical evaluation of model deficiencies.

Many sources of information can be used to reconstruct past climates. Pollen-based
reconstructions are the most widespread, and pollen-based data were the basis for the
current standard LGM benchmark data set by Bartlein et al. (2011). In common with
other data sources, the pollen-based reconstructions were generated for individual sites.
Geological preservation issues mean that the number of sites available inevitably
decreases through time (Bradley, 2014). Since pollen is only preserved for a long time
in anoxic sediments, the geographic distribution of potential sites is biased towards
climates that are relatively wet today. Furthermore, the actual sampling of potential
sites is highly non-uniform, so there are large geographic gaps in data coverage
(Harrison et al., 2016). The lack of continuous climate fields is not ideal for model
evaluation, and so attempts have been made to generalize the site-based data either
through gridding, interpolation, or some form of multiple regression (see e.g. Bartlein
et al., 2011; Annan and Hargreaves, 2013). However, there has so far been no attempt
to produce a physically consistent, multi-variable reconstruction which provides the
associated uncertainties explicitly.
A further characteristic of the LGM that creates problems for quantitative
reconstructions based on pollen data is the much lower atmospheric carbon dioxide
concentration, [$CO_2$], compared to the pre-industrial Holocene. [$CO_2$] has a direct effect
on plant physiological processes. Low [$CO_2$] as experienced by plants at the LGM is
expected to have led to reduced water-use efficiency – the ratio of carbon assimilation
to the water lost through transpiration (Bramley et al., 2013). Most reconstructions of
moisture variables from pollen data, including most of the reconstructions used by
Bartlein et al. (2011), do not take [$CO_2$] effects into account. Yet several modelling
studies have shown that the impact of low [$CO_2$] around the LGM on plant growth and
distribution was large (e.g. Jolly and Haxeltine, 1997; Cowling and Sykes, 1999;
Harrison and Prentice, 2003; Bragg et al., 2013; Martin Calvo et al., 2014; Martin Calvo
and Prentice, 2015). A few reconstructions of LGM climate based on the inversion of
process-based biogeography models have also shown large effects of low [$CO_2$] on
reconstructed LGM palaeoclimates (e.g. Guiot et al., 2000; Wu et al., 2007). The
reconstructions of moisture variables in the Bartlein et al. (2011) data set are thus
probably not reliable, and likely to be biased low.
Prentice et al. (2017) demonstrated an approach to correct reconstructions of moisture
variables for the effect of [$CO_2$], but this correction has not been applied globally. A
key side effect of applying this [$CO_2$] correction is to reconcile semi-quantitative
hydrological evidence for wet conditions at the LGM with the apparent dryness
suggested by the vegetation assemblages (Prentice et al., 2017). Similar considerations
apply to the interpretation of future climate changes in terms of vegetational effects.
Projections of future aridity (based on declining indices of moisture availability) linked
to warming are unrealistic, in a global perspective, because of the counteracting effect
of increased water use efficiency due to rising [$CO_2$] – which is generally taken into

account by process-based ecosystem models, but not by statistical models, using projected changes in vapour pressure deficit or some measure of plant-available water (Keenan et al., 2011; Roderick et al., 2015; Greve et al., 2017).

In this paper, we use variational data assimilation based on both pollen-based climate reconstructions and climate model outputs to arrive at a best-estimate analytical reconstruction of LGM climate, explicitly taking account of the impact of $[CO_2]$. Variational techniques provide a way of combining observations and model outputs to produce climate reconstructions that are not exclusively constrained to one source of information or the other (Nichols, 2010). We use the uncertainty contributions to the analytical reconstruction to provide confidence intervals for these reconstructions and also to delimit geographical regions for which the palaeodata provide no constraint on the reconstructions. The resulting data set is expected provide a well-founded multi-variable LGM climate dataset for palaeoclimate model benchmarking in CMIP6.

## 2 Methods

### 2.1 Pollen-based climate reconstructions

Bartlein et al. (2011) provided a global synthesis of pollen-based quantitative climate reconstructions for the LGM. The Bartlein et al. (2011) data set includes reconstructions of climate anomalies (differences between LGM and recent climates) for six variables (and their uncertainties), specifically mean annual temperature (MAT), mean temperature of the coldest month (MTCO), mean temperature of the warmest month (MTWA), growing degree days above a baseline of above 5°C (GDD5), mean annual precipitation (MAP), and an index of plant-available moisture (the ratio of actual to equilibrium evapotranspiration, or α). There are a small number of LGM sites (94) in the Bartlein et al. (2011) data set where model inversion was used to make the reconstructions of α and MAP;. no $[CO_2]$ correction is applied to these reconstructions. There are no data from Australia in the Bartlein et al. (2011) data set, and we therefore use quantitative reconstructions of MAT and another moisture index (MI), the ratio of MAP to potential evapotranspiration, from Prentice et al. (2017). Prentice et al. (2017) provide values of MI both before and after correction for $[CO_2]$; we use the uncorrected values in order to apply the correction for $[CO_2]$ within our assimilation framework. For consistency between the two data sets, we re-expressed reconstructions of α in terms of MI via the Fu-Zhang formulation of the Budyko relationship between actual evapotranspiration, potential evapotranspiration and precipitation (Zhang et al., 2004; Gallego-Sala et al., 2016).

The spatial coverage of the final data set is uneven (Figure 1). There are many more data points in Europe and North America than elsewhere. South America has the fewest

(14 sites). The number of variables available at each site varies: although most sites
(279) have reconstructions of at least three variables, some sites have reconstructions
of only one variable (60). Nevertheless, in regions where there is adequate coverage,
the reconstructed anomaly patterns are coherent, plausible and consistent among
variables.

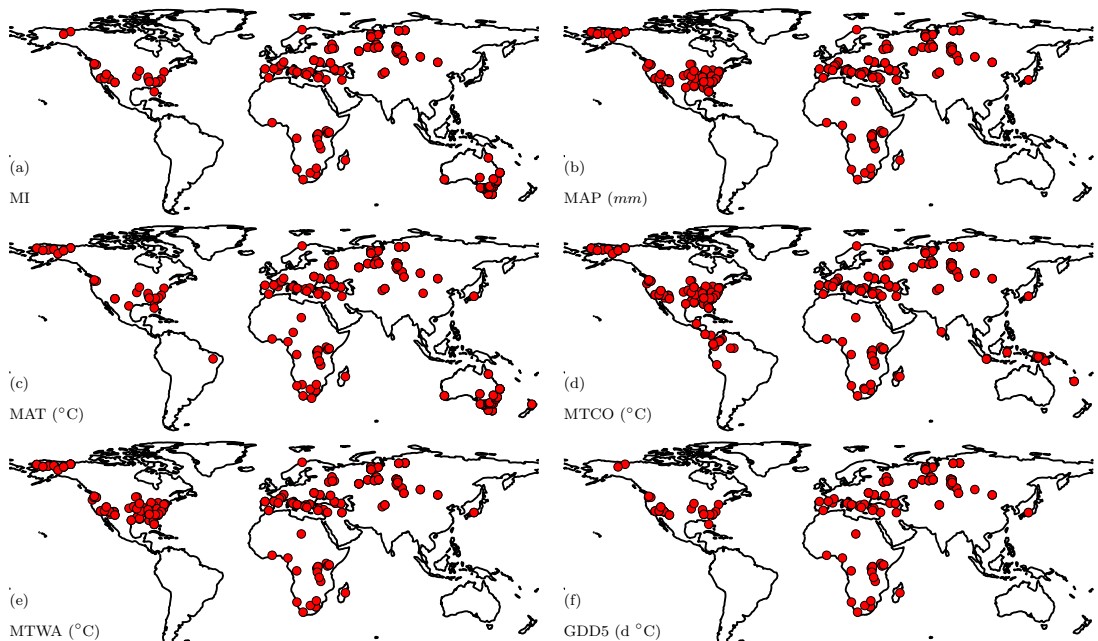

Figure 1: The distribution of the site-based reconstructions of climatic variables at the
Last Glacial Maximum. The individual plots show sites providing reconstructions of
(a) moisture index (MI), (b) mean annual precipitation (MAP), (c) mean annual
temperature (MAT), (d) mean temperature of the coldest month (MTCO), (e) mean
temperature of the warmest month (MTWA) and growing degree days above a baseline
of $5^\circ$ C (GDD5). The original reconstructions are from Bartlein et al. (2011) and
Prentice et al. (2017).

For this application, we derived absolute LGM climate reconstructions by adding the
reconstructed climate anomalies at each site to the modern climate values from the
Climate Research Unit (CRU) historical climatology data set (CRU CL v2.0 dataset,
New et al., 2002), which provides climatological averages of monthly temperature,
precipitation and cloud cover fraction for the period 1961-1990 CE. Most of the climate
variables (MTCO, MTWA, MAT, MAP) can be calculated directly from the CRU CL
v2.0 dataset. GDD5 was calculated from pseudo-daily data derived by linear
interpolation of the monthly temperatures. MI was calculated from the CRU climate
variables using the radiation calculations in the SPLASH model (Davis et al., 2017).
For numerical efficiency, we non-dimensionalised all of the absolute climate
reconstructions (and their standard errors) before applying the variational techniques
(for details, see Cleator et al., 2019a).

## 2.2 Climate model simulations

Eight LGM climate simulations (Table 1) from the third phase of the Palaeoclimate Modelling Intercomparison Project (PMIP3: Braconnot et al., 2012) were used to create a prior. The PMIP LGM simulations were forced by known changes in incoming solar radiation, changes in land-sea geography and the extent and location of ice sheets, and a reduction in $[CO_2]$ to 185 ppm (see Braconnot et al., 2012 for details of the modelling protocol). We used the last 100 years of each LGM simulation. We interpolated monthly precipitation, monthly temperature and monthly fraction of sunshine hours from each LGM simulation and its pre-industrial (PI) control to a common 2 x 2° grid. Simulated climate anomalies (LGM minus PI) for each grid cell were then added to modern climate values calculated from the CRU CL 2.0 data set (New et al., 2002), as described for the pollen-based reconstructions, to derive absolute climate values. We calculated the multi-model mean and variance (Figure 2) across the models for each of the climate variables to produce the gridded map used as the prior.

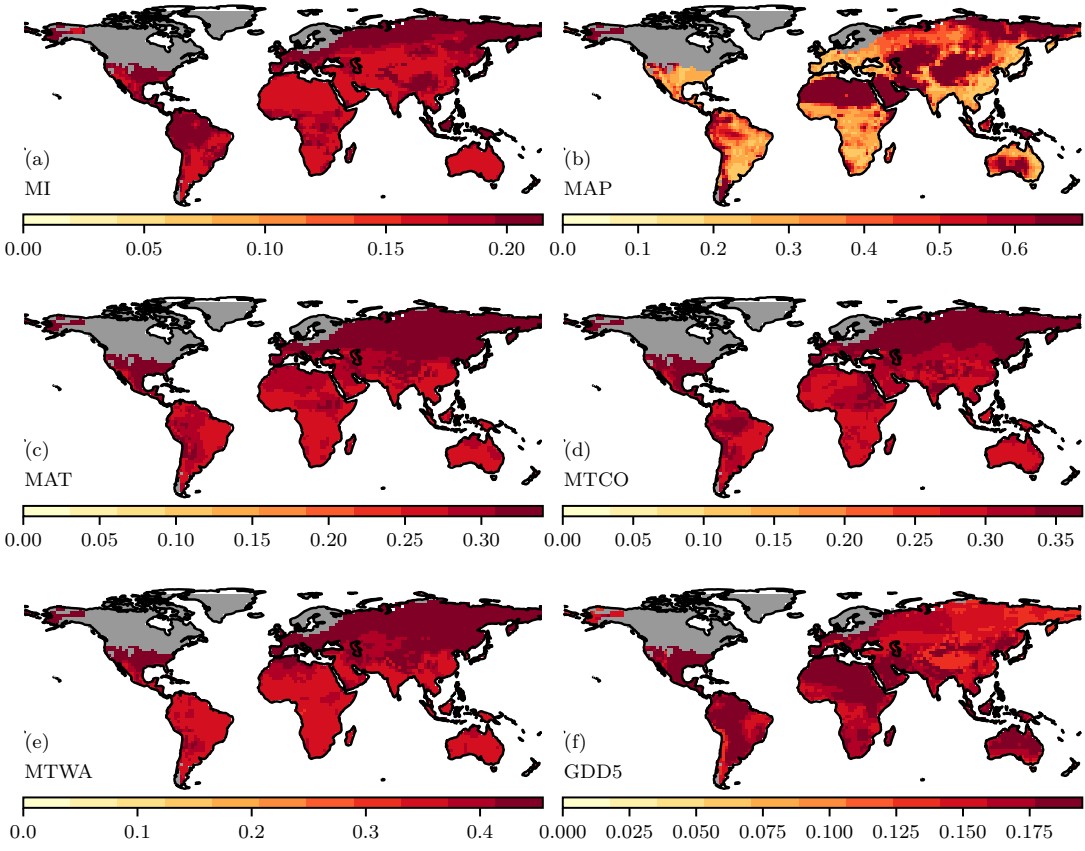

Figure 2: Uncertainties associated with the climate prior. The climate is derived from a multi-model mean of the ensemble of models from the Palaeoclimate Modelling Intercomparison Project (PMIP) and is shown in SI Figure 1. The uncertainties shown

here are the standard deviation of the multi-model ensemble values. The individual
plots show the variance for the simulated (a) moisture index (MI), (b) mean annual
precipitation (MAP), (c) mean annual temperature (MAT), (d) mean temperature of the
coldest month (MTCO), (e) mean temperature of the warmest month (MTWA) and
growing degree days above a baseline of 5∘ C (GDD5).
## 2.3 Water-use efficiency calculations
We applied the general approach developed by Prentice et al. (2017) to correct pollen-
based statistical reconstructions to account for [$CO_2$] effects. The approach as
implemented here is based on equations (Appendix 1) that link moisture index (MI) to
transpiration and the ratio of leaf-internal to ambient $CO_2$. The correction is based on
the principle that the rate of water loss per unit carbon gain is inversely related to
effective moisture availability as sensed by plants. The method involves solving a non-
linear equation that relates rate of water loss per unit carbon gain to MI, temperature
and $CO_2$ concentration. The equation is derived from theory that predicts the response
of the ratio of leaf-internal to ambient [$CO_2$] to vapour pressure deficit and temperature
(Prentice et al., 2014; Wang et al., 2014).
## 2.4 Application of variational techniques
Variational data assimilation techniques provide a way of combining observations
and model outputs to produce climate reconstructions that are not exclusively
constrained to one source of information or the other (Nichols, 2010). We use the
3D-variational method, described in Cleator et al. (2019a) to find the maximum a
posteriori estimate (or analytical reconstruction) of the palaeoclimate given the
site-based reconstructions and the model-based prior. The method constructs a
cost function, which describes how well a particular climate matches both the site-
based reconstructions and the prior, by assuming the reconstructions and prior
have a Gaussian distribution. To avoid sharp changes in time and/or space in the
analytical reconstructions, the method assumes that the prior temporal and
spatial covariance correlations are derived from a modified Bessel function, in
order to create a climate anomaly field that is smooth both from month to month
and from grid cell to grid cell. The degree of correlation is controlled through two
length scales: a spatial length scale that determines how correlated the covariance
in the prior is between different geographical areas, and a temporal length scale
that determines how correlated it is through the seasonal cycle. The site-based
reconstructions are assumed to have negligible correlations at these space and
time scales. The maximum a posteriori estimate is found by using the limited
memory Broyden- Fletcher-Goldfarb-Shanno method (Liu and Nocedal 1989) to
determine the climate that minimises the cost function. A first order estimate of
the analysis uncertainty covariance is also computed.
An observation operator based on calculations of the direct impact of [$CO_2$] on
water-use efficiency (section 2.3) is used in making the analytical reconstructions.
The prior is constructed as the average of eight LGM climate simulations (section
2.2). We use an ensemble of different model responses to the same forcing to
provide a series of physically consistent possible states, which can be viewed as
perturbed responses and provide the variance around the climatology provided
by the ensemble average. The prior uncertainty correlations are based on a
temporal length scale ($L_t$) of 1 month and a spatial length scale ($L_s$) of 400km.
Cleator et al., (2019a) have shown that a temporal length scale of 1 month
provides an adequately smooth solution for the seasonal cycle, both using single
sites and over multiple grid cells, as shown by the sensitivity of the resolution
matrix (Menke, 2012; Delahaies et al., 2017) to changes in the temporal length
scale. Consideration of the spatial spread of variance in the analytical
reconstruction shows that a spatial length scale of 400km also provides a
reasonable reflection of the large-scale coherence of regional climate change.
We generated composite variances on the analytical reconstructions (Figure 3) by
combining the covariances from the site-based reconstructions and from the
prior. There are regions where all of the models systematically differ from the site-
based reconstructions (Harrison et al., 2015) but nevertheless the inter-model
variability is low, which would lead to a very small contribution to the composite
uncertainties from the prior. We therefore calculated the uncertainty of the prior
from an equal combination of the global uncertainty, the average variance
between each grid cell, and local uncertainty, the variance between the different
models. The reliability of the analytical reconstructions was assessed by
comparing these composite covariances with the uncertainties in the prior. We
masked out cells where the inclusion of site-based reconstructions does not
produce an improvement of > 5% from the prior. Since this assessment is based
on a change in the variance, rather than absolute values, this masking removes
regions where there are no pollen-based reconstructions or the pollen-based
reconstructions have very large uncertainties.

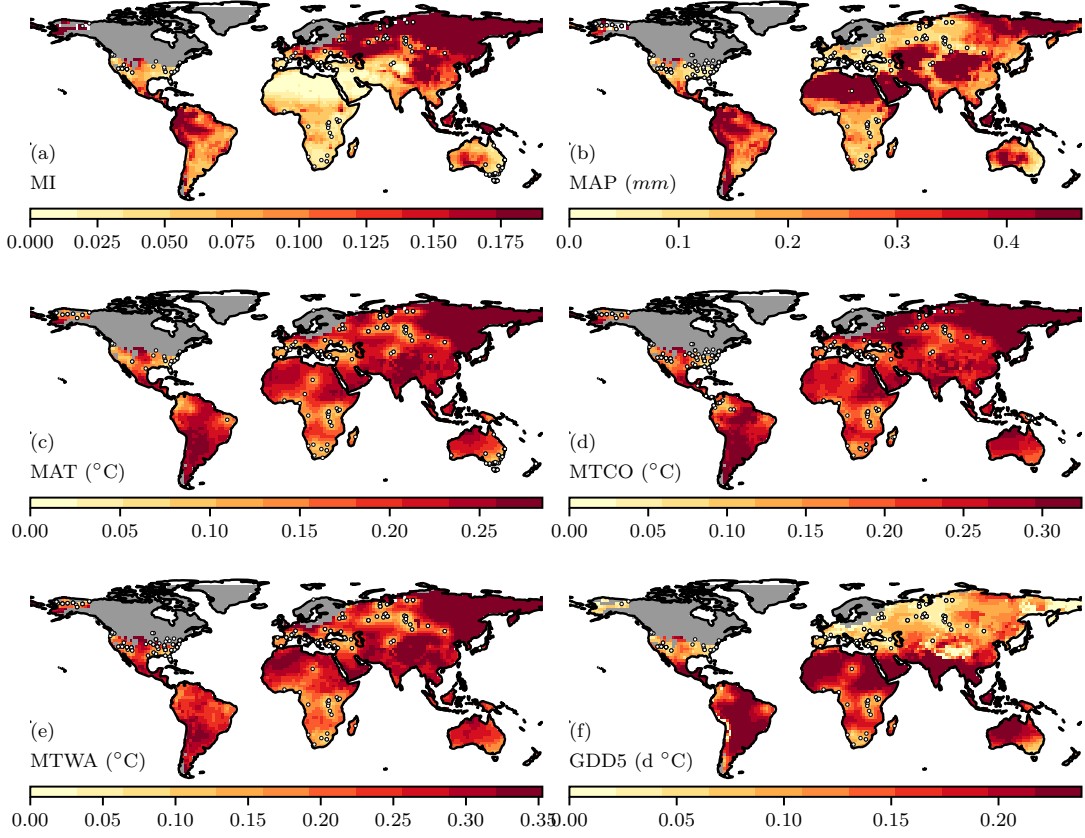

Figure 3: Uncertainties on the analytical reconstructions. These uncertainties represent a combination of the uncertainty on the site-based reconstructions, and the grid-based variance on the prior and the global variance from the prior.

## 3 Results

The analytical reconstructions (Figure 4) show an average year-round cooling of –5.6 °C in the northern extratropics. The cooling is larger in winter (–7.6 °C) than in summer (–2.4 °C). A limited number of grid cells in central Eurasia show warmer-than-present summers, and higher MAT. Temperature changes are more muted in the tropics, with an average change in MAT of –3.7 °C. The cooling is somewhat lower in summer than winter (–2.7 °C compared to –4.1 °C). Reconstructed temperature changes were slightly larger in the southern extratropics, with average changes in MAT of –5.0°C, largely driven by cooling in winter.

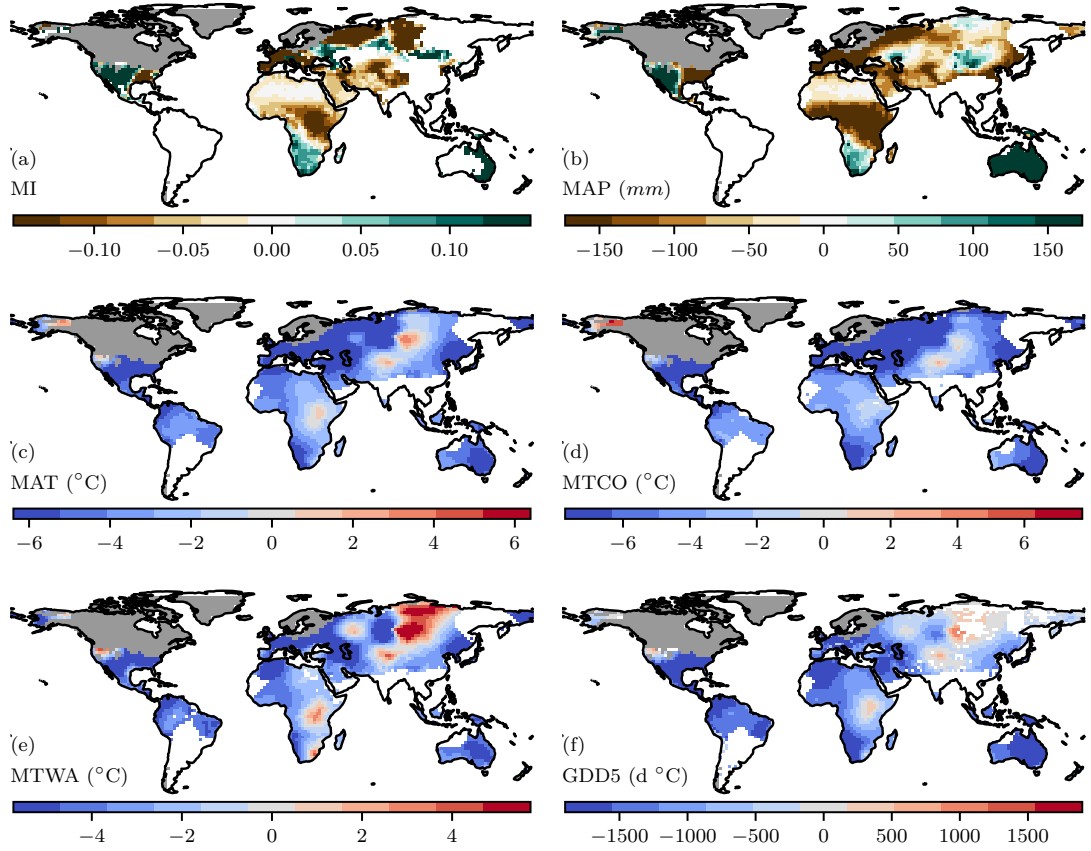

Figure 4: Analytically reconstructed climate, where areas for which the site-based data provide no constraint on the prior have been masked out. The individual plots show reconstructed (a) moisture index (MI), (b) mean annual precipitation (MAP), (c) mean annual temperature (MAT), (d) mean temperature of the coldest month (MTCO), (e) mean temperature of the warmest month (MTWA) and growing degree days above a baseline of 5∘ C (GDD5). The anomalies are expressed relative to the long term average (1960-1990) values from the Climate Research Unit (CRU) historical climatology data set (CRU CL v2.0 dataset, New et al., 2002).

Changes in moisture-related variables (MAP, MI) across the northern hemisphere are geographically more heterogeneous than temperature changes. Reconstructed MAP is greater than present in western North America (172 mm) but less than present (–29 mm) in eastern North America. Most of Europe is reconstructed as drier than present (–305mm), the same for eastern Eurasia (-94 mm) and the Far East (–66 mm). The patterns in MI are not identical to those in MAP, because of the influence of temperature on MI, but regional changes are generally similar to those shown by MAP. Most of the tropics are shown as drier than present while the southern hemisphere extratropics are wetter than present, in terms of both MAP and MI.

The reconstructed temperature patterns are not fundamentally different from those shown by Bartlein et al. (2011) but the analytical dataset provides information for a

much larger area (1153% increase) thanks to the method's imposition of consistency
among different climate variables, and smooth variations both in space and through the
seasonal cycle. There are systematic differences, however, between the analytical
reconstructions and the pollen-based reconstructions in terms of moisture-related
variables (MAP, MI) because the analytical reconstructions take account of the direct
influence of [CO2] on plant growth. The physiological impact of [CO2] leads to
analytical reconstructions indicating wetter than present conditions in many regions
(Figure 5a, Figure 5b), for example in southern Africa where several of the original
pollen-based reconstructions show no change in MAP or MI compared to present, but
the analytical reconstruction shows wetter conditions than present. In some regions,
incorporating the impact of [$CO_2$] reverses the sign of the reconstructed changes. Part
of northern Eurasia is reconstructed as being wetter than present, despite pollen-based
reconstructions indicating conditions drier than present (both in terms of MAP and MI),
as shown by SI Figure 3. The relative changes in MAP and MI are similar across all
sites (Figure 5c), implying that the analytically reconstructed changes are driven by
changes in precipitation rather than temperature.

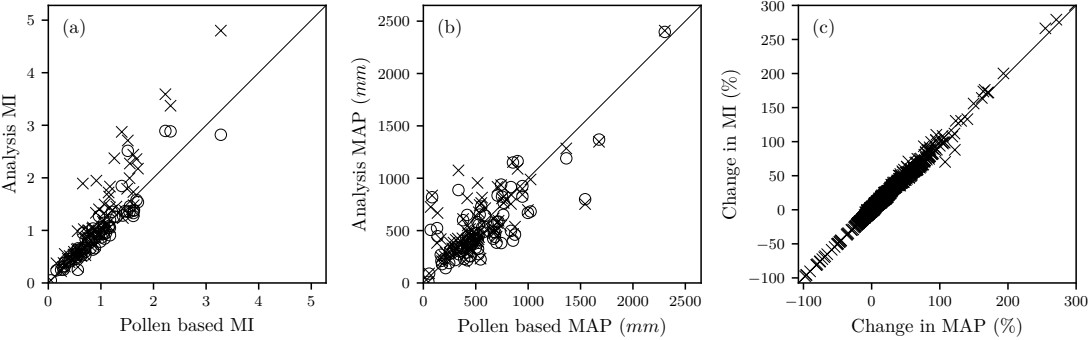


Figure 5: Impact of $CO_2$ on reconstructions of moisture-related variables. The
individual plots show (a) the change in moisture index (MI) and (b) the change in mean
annual precipitation (MAP) compared to the original pollen-based reconstructions for
the LGM before (circles) and after (crosses) the physiological impacts of [$CO_2$] on
water-use efficiency are taken into account. The third plot (c) shows the relative
difference in MI and MAP as a result of [$CO_2$], shown as the percentage difference
between the no-[$CO_2$] and [$CO_2$] calculations.

**4 Discussion**

Variational data assimilation techniques provide a way of combining observations and
model outputs, taking account the uncertainties in both, to produce a best-estimate
analytical reconstruction of LGM climate. These reconstructions extend the
information available from site-based reconstructions both spatially and through the
seasonal cycle. Our new analytical data set characterizes the seasonal cycle across a
much larger region of the globe than the data set that is currently being used for
benchmarking of palaeoclimate model simulations. We therefore suggest that this data
set (Cleator et al. 2019b) should be used for evaluating the CMIP6-PMIP4 LGM
simulations.
Some areas are still poorly covered by quantitative pollen-based reconstructions of
LGM climate, most notably South America. More pollen-based climate reconstructions
would provide one solution to this problem – and there are many pollen records that
could be used for this purpose (Flantua et al., 2015; Herbert and Harrison, 2016;
Harrison et al., 2016). There are also quantitative reconstructions of climate available
from individual sites (e.g. Lebamba et al., 2012; Wang et al., 2014; Loomis et al., 2017;
Camuera et al., 2019) that should be incorporated into future data syntheses.  It would
also be possible to incorporate other sources of quantitative information, such as
chironomid-based reconstructions (e.g. Chang et al., 2015), within the variational data
assimilation framework.
One of the benefits of the analytical framework applied here is that it allows the
influence of changes in $[CO_2]$ on the moisture reconstructions to be taken into account.
Low $[CO_2]$ must have reduced plant water-use efficiency, because at low $[CO_2]$ plants
need to keep stomata open for longer in order to capture sufficient $CO_2$. Statistical
reconstruction methods that use modern relationships between pollen assemblages and
climate under modern conditions (i.e. modern analogues, transfer functions, response
surfaces: see Bartlein et al., 2011 for discussion) cannot account for such effects.
Climate reconstruction methods based on the inversion of process-based ecosystem
models can do so (see e.g. Guiot et al., 2000; Wu et al., 2007; Wu et al., 2009; Izumi
and Bartlein, 2016) but are critically dependent on the reliability of the vegetation
model used. Most of the palaeoclimate reconstructions have been made by inverting
some version of the BIOME model (Kaplan et al., 2003), which makes use of
bioclimatic thresholds to separate different plant functional types (PFTs). As a result,
reconstructions made by inversion show "jumps" linked to shifts between vegetation
types dominated my different PFTs whereas, as has been shown recently (Wang et al.,
2017), differences in water use efficiency of different PFTs can be almost entirely
accounted for by a single equation, as proposed here. Sensitivity analyses show that the
numerical value of the corrected moisture variables (MI, MAP) is dependent on the
reconstructed values of these variables but is insensitive to uncertainties in the
temperature and moisture inputs (Prentice et al., 2017). The strength of the correction
is primarily sensitive to [CO2], but the LGM [CO2] value is well constrained from ice-
core records. The response of plants to changes in $[CO_2]$ is non-linear (Harrison and
Bartlein, 2012), and the effect of the change between recent and pre-industrial or mid-
Holocene conditions is less than that between pre-industrial and glacial conditions.
Nevertheless, it would be worth taking the [CO2] effect on water-use efficiency into
account in making reconstructions of interglacial time periods as well.

The influence of individual pollen-based reconstructions on the analytical
reconstruction of seasonal variability, or the geographic area influenced by an
individual site, is crucially dependent on the choice of length scales. We have adopted
conservative length scales of 1 month and 400 km, based on sensitivity experiments
made for southern Europe (Cleator et al., 2019a). These length scales produce
numerically stable results for the LGM, and the paucity of data for many regions at the
LGM means that using fixed, conservative length scales is likely to be the only practical
approach. However, in so far as the spatial length scale is related to atmospheric
circulation patterns, there is no reason to suppose that the optimal spatial length scale
will be the same from region to region. The density and clustering of pollen-based
reconstructions could also have a substantial effect on the optimal spatial length scale.
A fixed 1-month temporal length scale is appropriate for climates that have a reasonably
smooth and well-defined seasonal cycle, either in temperature or precipitation.
However, in climates where the seasonal cycle is less well defined, for example in the
wet tropics, or in situations where there is considerable variability on sub-monthly time
scales, other choices might be more appropriate. For time periods such as the mid-
Holocene, which have an order of magnitude more site-based data, it could be useful to
explore the possibilities of variable length scales.
We have used a 5% reduction in the analytical uncertainty compared to prior
uncertainty to identify regions where the incorporation of site-based data has a
negligible effect on the prior as a way of masking out regions for which the observations
have effectively no impact on the analytical reconstructions. The choice of a 5% cut-
off is arbitrary, but little would be gained by imposing a more stringent cut-off at the
LGM given that many regions are represented by few observations. A more stringent
cut-off could be applied for other time intervals with more data. We avoid the use of a
criterion based on the analytical reconstruction showing any improvement on the prior
because this could be affected by numerical noise in the computation. Alternative
criteria for the choice of cut-off could be based on whether the analytical reconstruction
had a reduced uncertainty compared to the pollen-based reconstructions or could be
derived by a consideration of the condition number used to select appropriate length
scales.
There have been a few previous attempts to use data assimilation techniques to generate
spatially continuous palaeoclimate reconstructions. Annan and Hargreaves (2013) used
a similar multi-model ensemble as the prior and the pollen-based reconstructions from
Bartlein et al. (2011) to reconstruct MAT at the LGM. However, they made no attempt
to reconstruct other seasonal variables, either independently, or through exploiting
features of the simulations (as we have done here) to generate seasonal reconstructions.
Particle filter approaches (e.g. Goosse et al., 2006; Dubinkina et al., 2011) produce
dynamic estimates of palaeoclimate, but particle filters cannot produce estimates of
climate outside the realm of the model simulations. Our 3-D variational data

assimilation approach has the great merit of being able to produce seasonally coherent reconstructions generalized over space, while at the same time being capable of producing reconstructions that are outside those captured by the climate model, because they are not constrained by a specific source (Nichols, 2010). This property is of particular importance if the resulting data set is to be used for climate model evaluation, as we propose.

**Data availability.** The gridded data for the LGM reconstructions are available from DOI:10.17864/1947.229; the code used to generate these reconstructions is available from DOI:10.5281/zenodo.3445166.

**Author Contributions**

All authors contributed to the design of the study; ICP developed the theory underlying the $CO_2$ correction; SC implemented the analyses. SC and SPH wrote the first version of the manuscript, and all authors contributed to the final version.

**Competing Interests.**

The authors declare they have no competing interests.

**Acknowledgements**

SFC was supported by a UK Natural Environment Research Programme (NERC) scholarship as part of the SCENARIO Doctoral Training Partnership. SPH acknowledges support from the ERC-funded project GC 2.0 (Global Change 2.0: Unlocking the past for a clearer future, grant number 694481). ICP acknowledges support from the ERC under the European Union's Horizon 2020 research and innovation programme (grant agreement no: 787203 REALM) This research is a contribution to the AXA Chair Programme in Biosphere and Climate Impacts and the Imperial College initiative on Grand Challenges in Ecosystems and the Environment (ICP). NKN is supported in part by the NERC National Center for Earth Observation (NCEO). We thank PMIP colleagues who contributed to the production of the palaeoclimate reconstructions. We also acknowledge the World Climate Research Programme's Working Group on Coupled Modelling, which is responsible for CMIP, and the climate modelling groups in the Paleoclimate Modelling Intercomparison Project (PMIP) for producing and making available their model output. For CMIP, the U.S. Department of Energy's Program for Climate Model Diagnosis and Intercomparison provides coordinating support and led development of software infrastructure in partnership with the Global Organization for Earth System Science Portals. The analyses and figures are based on data archived at CMIP on 12/09/18.

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

**Figures and Tables Captions**

Figure 1: The distribution of the site-based reconstructions of climatic variables at the Last Glacial Maximum. The individual plots show sites providing reconstructions of (a) moisture index (MI), (b) mean annual precipitation (MAP), (c) mean annual temperature (MAT), (d) mean temperature of the coldest month (MTCO), (e) mean temperature of the warmest month (MTWA) and growing degree days above a baseline of $5\circ$ C (GDD5). The original reconstructions are from Bartlein et al. (2011) and Prentice et al. (2017).

Figure 2: Uncertainties associated with the climate prior. The climate is derived from a multi-model mean of the ensemble of models from the Palaeoclimate Modelling Intercomparison Project (PMIP) and is shown in SI Figure 1. The uncertainties shown here are the standard deviation of the multi-model ensemble values. The individual plots show the variance for the simulated (a) moisture index (MI), (b) mean annual precipitation (MAP), (c) mean annual temperature (MAT), (d) mean temperature of the coldest month (MTCO), (e) mean temperature of the warmest month (MTWA) and growing degree days above a baseline of $5\circ$ C (GDD5).

Figure 3: Uncertainties on the analytical reconstructions. These uncertainties represent a combination of the uncertainty on the site-based reconstructions, and the grid-based variance on the prior and the global variance from the prior.

Figure 4: Analytically reconstructed climate, where areas for which the site-based data provide no constraint on the prior have been masked out. The individual plots show reconstructed (a) moisture index (MI), (b) mean annual precipitation (MAP), (c) mean annual temperature (MAT), (d) mean temperature of the coldest month (MTCO), (e) mean temperature of the warmest month (MTWA) and growing degree days above a baseline of $5\circ$ C (GDD5). The anomalies are expressed relative to the long term average (1960-1990) values from the Climate Research Unit (CRU) historical climatology data set (CRU CL v2.0 dataset, New et al., 2002).

Figure 5: Impact of $CO_2$ on reconstructions of moisture-related variables. The individual plots show (a) the change in moisture index (MI) and (b) the change in mean annual precipitation (MAP) compared to the original pollen-based reconstructions for the LGM before (circles) and after (crosses) the physiological impacts of $[CO_2]$ on water-use efficiency are taken into account. The third plot (c) shows the relative difference in MI and MAP as a result of $[CO_2]$, shown as the percentage difference between the no-$[CO_2]$ and $[CO_2]$ calculations.

Table 1: Details of the models from the Palaeoclimate Modelling Intercomparison
Project (PMIP) that were used for the Last Glacial Maximum (LGM) simulations used
to create the prior.
Table 1: *Details of the models from the third phase of the Palaeoclimate Modelling*
*Intercomparison Project (PMIP3) that were used for the Last Glacial Maximum*
*(LGM) simulations used to create the prior. Coupled ocean-atmosphere models are*
*indicated as OA, which OAC models have a fully interactive carbon cycle. The*
*resolution in the atmospheric, oceanic and sea ice components of the models is given*
*in terms of numbers of grid cells in latitude and longitude.*

| Model name | Type | Resolution | | | Year length | Reference |
|---|---|---|---|---|---|---|
| | | Atmosphere | Ocean | Sea Ice | | |
| CCSM4 | OA | 192, 288 | 320, 384 | 320, 384 | 365 | Gent et al. (2011) |
| CNRM-CM5 | OA | 128, 256 | 292, 362 | 292, 362 | 365-366 | Voldoire et al. (2012) |
| MPI-ESM-P | OA | 96, 192 | 220, 256 | 220, 256 | 365-366 | Jungclaus et al. (2006) |
| MRI-CGCM3 | OA | 160, 320 | 360, 368 | 360, 368 | 365 | Yukimoto et al. (2011) |
| FGOALS-g2 | OA | 64, 128 | 64, 128 | 64, 128 | 365 | Li et al. (2013) |
| COSMOS-ASO | OAC | 96, 48 | 120, 101 | 120, 101 | 360 | Budich et al. (2010) |
| IPSL-CM5A-LR | OAC | 96, 96 | 149, 182 | 149, 182 | 365 | Dufresne et al., 2013 |
| MIROC-ESM | OAC | 64, 128 | 192, 256 | 192, 256 | 365 | Watanabe et al. (2011) |


**Appendix**

We define $e$ as the water lost by transpiration ($E$) per unit carbon gained by photosynthesis ($A$). This term, the inverse of the water use efficiency, is given by:

$$e = E/A = 1.6\, D\, / ((1 - \chi)\, c_a) \tag{A1}$$

where $D$ is the leaf-to-air vapour pressure deficit (Pa), $c_a$ is the ambient $CO_2$ partial pressure (Pa) and $\chi$ is the ratio of leaf-internal $CO_2$ partial pressure ($c_i$) to $c_a$. An optimality-based model (Prentice *et al.* 2014), which accurately reproduces global patterns of $\chi$ and its environmental dependencies inferred from leaf $\delta^{13}C$ measurements (Wang *et al.* 2017), predicts that:

$$\chi = (\Gamma^*/c_a) + (1 - \Gamma^*/c_a)\, \xi/(\xi + \sqrt{D}) \tag{A2a}$$

and

$$\xi = \sqrt{(\beta(K + \Gamma^*)/1.6\, \eta^*)} \tag{A2b}$$

where $\Gamma^*$ is the photorespiratory compensation point of $C_3$ photosynthesis (Pa), $\beta$ is a constant (estimated as 240 by Wang *et al.* 2017), $K$ is the effective Michaelis-Menten coefficient of Rubisco (Pa), and $\eta^*$ is the ratio of the viscosity of water (Pa s) at ambient temperature to its value at 25˚C. Here $K$ depends on the Michaelis-Menten coefficients of Rubisco for carboxylation ($K_C$) and oxygenation ($K_O$), and on the partial pressure of oxygen $O$ (Farquhar *et al.* 1980):

$$K = K_C\, (1 + O/K_O) \tag{A3}$$

Standard values and temperature dependencies of $K_C$, $K_O$, $\Gamma^*$ and $\eta^*$ are assigned as in Wang *et al.* (2017).

The moisture index MI is expressed as

$$MI = P/E_q, \quad E_q = \sum_n (R_n/\lambda)\, s/(s + \gamma) \tag{A4}$$

where $P$ is annual precipitation, $R_n$ is net radiation for month n, $\lambda$ is the latent heat of vaporization of water, $s$ is the derivative of the saturated vapour pressure of water with respect to temperature (obtained from a standard empirical formula also used by Wang *et al.* 2017), and $\gamma$ is the psychrometer constant. We assume that values of MI reconstructed from fossil pollen assemblages, using contemporary pollen and climate data either in a statistical calibration method or in a modern-analogue search, need to be corrected in such a way as to preserve the contemporary relationship between MI and $e$, while taking into account the change in $e$ that is caused by varying $c_a$ and temperature away from contemporary values. The sequence of calculations is as follows. (1) Estimate $e$ and its derivative with respect to temperature ($\partial e/\partial T$) for the contemporary $c_a$ and climate, using equations (A1) – (A3) above. (2) Use the $e$ and $\partial e/\partial T$ *to c*alculate $\partial D/\partial T$ given the palaeo $c_a$ (measured in ice-core data) and temperature (reconstructed from pollen data), via a series of analytical equations that relate $\partial e/\partial T$ to $\partial D/\partial T$ and hence to $s$. (3) Use the new $\partial D/\partial T$ *and relative humidity (from the PMIP3 average)* to derive a new value of $s$. (4) Re-calculate MI using a palaeo estimate of $R_n$ (modelled as in Davis et al., 2017) and the new value of $s$.