# Peer review of "A new multi-variable benchmark for Last Glacial Maximum climate simulations"

_Climate of the Past, 2019_

## Referee Comment (RC1) · Anonymous Referee #1 · 1 Jul 2019

This paper uses quantitative estimates for climate parameters derived from pollen spectra from a number of previously published compilation studies for the Last Glacial Maximum (LGM). It combines these estimates with climate model output and a new approach to determining the impact of lower LGM CO2 on plant water use efficiency (as LGM CO2 was lower than today). The latter step is important because it changes the estimate of moisture availability from that which results from simply applying the pollen transfer function approach, because the transfer functions are effectively 'calibrated' for pre-industrial CO2 levels. The output from the study is a new global map of changes in a number of temperature and moisture parameters at the LGM compared to pre-industrial conditions. The study suggests that some parts of the world previously considered 'direr' based on pollen assemblages, may well have been 'wetter' when the

CO2 effect is considered. The approach also delineates the (large) areas of the planet where there is insufficient observational data to make an estimate of LGM conditions.

This is a relatively short paper that relies heavily on previously published work for data and methodology, which makes it hard to review in a very detailed fashion, because there is not much data included. The results presented are simply a number of global maps, with very little data in the text or supplementary information. As the previous studies have passed peer review this does not bother me too much, and the paper is clearly and well written. Hence I am happy to recommend publication with minor technical corrections.

A couple of points to consider: The definition for the LGM given here is 21±1 ka, and this appears to be because previous work has used this temporal extent. However it is different from, for example, to the range used by Annan and Hargreaves (cited in the paper) of 21±2ka, and recent work on sea level (Ishiwa et al. 2019) suggest the 'real' LGM was 19.1-19.7 ka, with a plateau prior from 20.4-25.9ka, both pushing out past the time interval use in this study. I don't think there is anything in particular to be done about this – just to think about. . .

Ishiwa, T., Yokoyama, Y., Okuno, J.I., Obrochta, S., Uehara, K., Ikehara, M. and Miyairi, Y., 2019. A sea-level plateau preceding the Marine Isotope Stage 2 minima revealed by Australian sediments. Scientific reports, 9(1), p.6449.

Again, a general point I guess, that the paper refers to several new studies since the Bartlein paper on which the analysis is based, and there are more. It would be nice to think these could be assimilated into a future dataset to maybe close some of the large 'no data' holes in the results. . .

Small things (in fact a laudably small number of small things): L59: change to 'lower, atmospheric aerosol. . .' L321: comma after 'however' (I think?)

---

## Short Comment (SC1) · 3 Jul 2019

We thank the reviewer for their positive and helpful comments on our manuscript.

1) As the reviewer points out, the pollen-based reconstructions and the climate model simulations underpinning our reconstruction are in the public domain, and the data assimilation methodology is described in detail in another publication. The general approach used for the CO2 corrections, which the reviewer describes as a significant contribution, was published in Prentice et al. (2017) – although we provide the equations for the implementation of this approach in the current paper in Appendix 1. Therefore, the new results in this paper are indeed the global maps of reconstructed climate variables. These data are archived and will be made publicly available – however, we

realise that it may not have been obvious that the citation to Cleator et al. (2019b) represented the reconstruction data set. We propose to modify the last sentence of the abstract to make it clear that the reconstruction data are available as follows:

The new reconstructions will provide a robust benchmark for evaluation of the PMIP4/CMIP6 entry-card LGM simulations and are available at DOI:10.17864/1947.206

We did not include a Data Availability section in the Discussion paper and we will also rectify this:

Data availability. The analytical reconstructions are available at the University of Reading repository, DOI:10.17864/1947.206.

2) The reviewer indicates that the definition of the LGM used in our paper (21±1 ka) differs from the interval used by Annan and Hargreaves of 21±2ka, and there is recent work on sea level (Ishiwa et al. 2019) which suggests the 'real' LGM was 19.1-19.7 ka, with a plateau prior to this from 20.4-25.9ka. Our choice of this time interval reflects the fact that the LGM is conventionally defined in PMIP at 21 ka and most of the pollen-based reconstructions of this interval included in the Bartlein et al data set are from the 21±1 ka. We are aware that there is still controversy over the timing of the LGM, with both younger and older ages mooted for the actual maximum ice volume/sea-level lowering (see e.g. Peltier and Fairbanks, 2006; Clark et al., 2009; Lambeck et al., 2014). Even the recent work by Ishiwa et al. (2019) points out that the sea level drop after 19.7 ka was only 10m and that there was a long plateau with stable low sea level prior to this and encompassing the 21 ka interval. Since our aim is to produce a data set for benchmarking new PMIP LGM simulations, which will be run with boundary conditions for 21 ka (Kageyama et al., 2017), the exact date of the LGM is therefore not an issue. However, we agree that there is a difference between the true definition of the LGM and the convention used for modelling purposes, and that this is not clear from our introductory text, so we propose to expand our definition (lines 57-61) as follows:

At the Last Glacial Maximum (LGM, conventionally defined for modelling purposes as 21 000 years ago), insolation was quite similar to the present, but global ice volume was at a maximum, eustatic sea level was close to a minimum, long-lived greenhouse gas concentrations were lower and atmospheric aerosol loadings higher than today, and land surface characteristics (including vegetation distribution) were also substantially different from today.

3) The reviewer points out that we refer to several new studies since the Bartlein paper on which the analysis is based, and there are more, and that it would be nice to think these could be assimilated into a future dataset to maybe close some of the large 'no data' holes in the results. We thoroughly agree that it would be good to plug the gaps, and this will be an effort for the future. The three papers that we cite at lines 361-363 (Flantua et al., 2015; Herbert and Harrison, 2016; Harrison et al., 2016) demonstrate that there are pollen records available that would plug the gaps, but alas do not provide quantitative reconstructions at these sites. The Izumi and Bartlein, 2016 paper provides an inversion-based reconstruction for North American – this region is already relatively well covered in the Bartlein et al data set. Similarly Mauri et al., 2015 provide a new gridded reconstruction for Europe – again a region that is well covered in the Bartlein et al data set. However, we are aware of new pollen-based quantitative reconstructions embracing the LGM for individual sites (e.g. in Africa, China, Russia, southern Europe) and compiling these reconstructions would certainly be a worthwhile effort in the future. Our method also lends itself to combining pollen-based reconstructions with other quantitative estimates of terrestrial palaeoclimate, and again this should be something that is done in the future. We will expand the paragraph describing future possibilities to expand the current data set to spell out some of these opportunities more clearly, as follows:

Some areas are still poorly covered by quantitative pollen-based reconstructions of LGM climate, most notably South America. More pollen-based climate reconstructions would provide one solution to this problem – and there are many pollen records

that could be used for this purpose (Flantua et al., 2015; Herbert and Harrison, 2016; Harrison et al., 2016). There are also quantitative reconstructions of climate available from individual sites (e.g. Lebamba et al., 2012; Wang et al., 2014; Loomis et al., 2017; Camuera et al., 2019) that should be incorporated into future data syntheses. It would also be possible to incorporate other sources of quantitative information, such as chironomid-based reconstructions (e.g. Chang et al., 2015) within the variational data assimilation framework.

Additional references Camuera, J., Jiménez-Moreno, G., Ramos-Román, M.J., García-Alix, A., Toney, J.L., Anderson, R.S., Jiménez-Espejo, F., Bright, J., Webster, C., Yanes, Y., José S. Carrión, J.S., 2019. Vegetation and climate changes during the last two glacial-interglacial cycles in the western Mediterranean: A new long pollen record from Padul (southern Iberian Peninsula), Quaternary Science Reviews, 205, 86-105, https://doi.org/10.1016/j.quascirev.2018.12.013. Chang, J.C., Shulmeister, J., Woodward, C., Steinberger, L., Tibby, J., Cameron Barr, C., 2015. A chironomid-inferred summer temperature reconstruction from subtropical Australia during the last glacial maximum (LGM) and the last deglaciation, Quaternary Science Reviews, 122, 282-292, https://doi.org/10.1016/j.quascirev.2015.06.006. Lebamba, J., Vincens, A., and Maley, J.: Pollen, vegetation change and climate at Lake Barombi Mbo (Cameroon) during the last ca. 33 000 cal yr BP: a numerical approach, Clim. Past, 8, 59-78, https://doi.org/10.5194/cp-8-59-2012, 2012. Loomis, S. E., Russell, J. M., Verschuren, D., Morrill, C., De Cort, G., Sinninghe Damsté, J. S., . . . Kelly, M. A. (2017). The tropical lapse rate steepened during the Last Glacial Maximum. Science advances, 3(1), e1600815. doi:10.1126/sciadv.1600815 Wang, Y., Herzschuh, U., Shumilovskikh, L. S., Mischke, S., Birks, H. J. B., Wischnewski, J., Böhner, J., Schlütz, F., Lehmkuhl, F., Diekmann, B., Wünnemann, B., and Zhang, C.: Quantitative reconstruction of precipitation changes on the NE Tibetan Plateau since the Last Glacial Maximum – extending the concept of pollen source area to pollen-based climate reconstructions from large lakes, Clim. Past, 10, 21-39, https://doi.org/10.5194/cp-10-21-2014, 2014.

L59: change to 'lower, atmospheric aerosol. . .' We will make this change.

L321: comma after 'however' We will make this change.

---

## Referee Comment (RC2) · Michel Crucifix (Referee) · 8 Aug 2019

The contribution provides up-to-date global maps of seasonal climatic indicators of the LGM, intended to be used as a reference ("benchmark") for PMIP4/CMIP6 entry-card LGM simulations. The reconstructions are obtained by a variational method using a prior based on the CMIP3/PMIP5 ensemble, and updated on pollen-based reconstructions ($CO_2$-corrected) of various indices.

The purpose of the study is clear, and the contribution is justified in the framework of the PMIP4 effort. However, before publication, it is advisable to revise the description of the methods and improve the wording accuracy.

[Figure]

**1 On the variational method**

I will focus here mainly on the variational technique (section 2.4). Mathematical details of the technique applied in this study are available in Cleator et al., 2019a. This is an arXiv preprint. It is not clear whether the latter is intended for a peer-reviewed journal or whether it was part of a thesis examination. I understand from that arXiv contribution that the different indices ($\alpha$, MAT, etc.) are first computed in the different simulations; precipitation is log-transformed to avoid negative predictions, and a matrix $B$ encodes the covariances between the indices simulated in the ensemble (this matrix is then called the covariance of the "uncertainties in the background"). The matrix $B$ is assumed to be the prior covariance.The variational approach further assumes Gaussian distributions and a fixed spatiotemporal covariance (with length scales of 1 month and 400 km, respectively). It is mathematically equivalent to Bayesian updating of a prior (the PMIP3/CMIP5 ensemble) by observations, which have their own error variances.

There are a number of points in this approach which deserve discussion. For this reason it would have been better to see these method details in the *Climate of the Past* paper, so that the paper, the review, and possible responses constitute a self-contained contribution.

1. Even though this is a reasonable and convenient choice, the PMIP3/CMIP5 ensemble is not a fully legitimate prior. For two reasons.

    1. Unlike what (roughly) obtains when using time series of a numerical weather prediction system, there is a priori no guarantee that the covariance matrix of a multi-model ensemble produces modes which satisfy "physical consistency". Why would we expect that the inter-model differences provide knowledge about how different variables should co-vary?

2. In principle, a "prior" encodes what we a priori believe the climate could be. The authors have then chosen to mask regions with little update by observations, and leave visible the grid points where the observations have seriously shifted the prior. This seems at first sight reasonable because the idea is to focus on the pollen reconstructions and not on the PMIP3 output. Yet, at face value, this approach is inconsistent with a Bayesian interpretation. Grid points of strong update are associated, in the Bayesian interpretation, with a very small marginal likelihood (a wrong prior means a wrong model). Hence, this leads me to two sub-questions:

   • To what extent should we be concerned that the posterior variance remains influenced by the prior variance? Indeed, mathematically, the posterior variance is bounded by the prior variance, which — if we admit the models are really off — is meaningless.

   • To what extent the prior covariance (link between different variables) may still be trusted at all if the models are so wrong? This remark strengthens the original concern about the physical meaning of the covariance matrix, even when the prior is only mildly updated. What is the advantage of this approach over a mere Gaussian interpolation (flat climate prior), which in this case might turn out to be more reliable and free of the dubious claim of "physical consistency"?

2. Were the length scales tested by some form of cross-validation (e.g. leave-one-out), or were they merely chosen because they are a priori reasonable?

3. The arXiv paper provides the definition of the moisture index. It should be repeated here (moisture index is currently introduced l. 297 without definition)

4. The authors should consider providing a link to supporting code. The maps are currently provided as University of Reading dataset (with a doi) but its lifecycle is detached from the present contribution. A dataset consistent with the current

*Climate of the Past* submission, reflecting a possible response to concerns of the reviewers, might best be included as supplementary information.

**2   Uncertainty (Uncertainties) vs variance**

- It is important to distinguish the notion of variance from the notion of uncertainty. They are not synonymous. Variance describes the second momentum of a distribution; uncertainty is a reference to an identified lack of knowledge. Only when the distribution is assumed reflects our knowledge of a given quantity is it legitimate to identify both. Multi-model ensembles, in general, cannot be said to capture our knowledge of the state of climate at a given time. For this reason, I would argue not to call the PMIP3 covariance a "background uncertainty". The legend of Figure 2 clearly identifies "uncertainties" with "standard deviation of the non-dimensionalised multi-model ensemble" but this seems inadequate to me. Adding to the confusion, different qualifiers occur throughout the text: "explicit uncertainty" (l. 97), "analytical uncertainty" (l. 406), and, on Figure 3, "grid-based errors in the prior" and "global uncertainty".

- As the uncertainty quantification seems to be a selling point of the present article, the assessment should be more open and transparent about sources of uncertainty, and discuss which of theses sources can be quantified and how. For example, little is said about uncertainties introduced by the $CO_2$ physiological correction. Is it guaranteed to be accurate?

- The strategy for identifying grid points with little posterior update explained l. 406 is not quite clear. Why not consider a Kullback-Leibler divergence? At the risk of repeating myself, I am concerned about the (meaningless) residual influence of the prior variance and covariance in cases where the prior is effectively discarded by the observations.

**3 About the discussion section**

- This is a minor comment, but the comparison with Goosse et al. 2006 is perhaps slightly misleading. The Goosse et al. purpose was dynamic reconstruction, while the purpose of the present contribution is to provide a diagnostic reconstruction. In passing, Goosse (2006) did not use a "Kalman particle filter" (whatever it means). Goosse et al. used what they called an "optimal realisation" iteration, which can be interpretated as a highly degenerate form of particle filter. Dubinkina et al. 2011, doi 10.1142/S0218127411030763, adopted a more standard particle filter.

- This said, the argument that the variational approach produces maps outside the realm of climate simulations is a double-edged sword. The variational approach assumes Gaussian distributions, and is mathematically equivalent to a Laplace approximation of arbitrary distributions. This is this approximation which allows generating posterior distributions far from the prior. But, in this case, sound Bayesian interpretation should lead us to treat such posterior as utterly suspicious.

- line 384 : It is said that it "would be worth taking [changes in length scales] into account." I would advise either deleting this sentence, or giving more substance to the claim. For example, have you already performed some sensitivity experiments.

**4 Other editorial comments**

- Is the very first paragraph really necessary?

- There is room for improving wording accuracy. In what sense is the benchmark

"robust" (l. 37) ? l. 97: You write: "explicit uncertainties attached to it". Did you mean "uncertainties explicitly attached" ? Avoid, where possible, the phrase "in terms of" or "means that" (ll. 321 - 326, in particular, need rewording). What is meant by a "statistical reconstruction method" l. 370 (the present exercise is a statistical reconstruction isnt'it ?).

- Figure 5: Shouldn't "pre-industrial reference" be preferred over the vague wording "original" as x-axis label?

---

## Referee Comment (RC3) · Anonymous Referee #3 · 28 Aug 2019

This paper demonstrates a valuable new approach to providing quantitative climate reconstructions based on pollen. This will be very useful for model-data comparisons in CMIP6/PMIP4 and beyond. The main advances here are the consistent and transparent correction for the effects of low atmospheric CO2 on plant moisture use, and the use of a statistical methodology to generate uncertainties and to interpolate spatially and seasonally.

The text is very well written and the figures are clear. However, the paper is quite short and lacks any detailed evaluation of the resultant product. The community's use of this new data product would in my opinion be aided by a more in-depth evaluation of the properties of the reconstruction. It's not clear how important the choices around the assimilation formulation are for the final reconstruction. Specifically the section around

lines 268-278 should in my opinion be spelled out and the sensitivity to these choices evaluated.

The statistical methodology that forms the basis of this study is also not described here but in a arXiv article. I'd like to see more of this brought into the present manuscript to make it self-contained.

Technical comments

Line 127: define MI here.

Line 209: modelsfor -> models for

Line 252-253: I think it might be appropriate to bring some/all of this methodology into the present text, as discussed above.

One question that arises from briefly reading the methodology paper, relates to figure 1 in the arXiv article. Here the assimilation appears not satisfy the pollen-inferred MTCO. Is this because the prior (from the models) is relatively consistent, and so doesn't allow the assimilation to get that cold? Does this happen when applied to the pollen data here? How do we interpret these choices, given that the climate models themselves could feasibly be systematically biased, e.g. through not including aerosols, or using modern vegetation distributions? How have you addressed the possible systematic bias in the models and hence in your prior?

Line 268-276: This section seems crucial to me, but is not clearly described. Please include the mathematical formulation used and a justification for choices made.

Lines 276-278: Do you mean that if the data is too uncertain you mask it based on a 5% criteria? Please could you re-phrase to clarify.

Lines 288: How does your product compare with the original Bartlein et al 2011, and the GCM-based prior? Could you show this?

How well is the seasonality captured and how does it differ from the simulated seasonality in the GCM prior?

---

## Editor Comment (EC1) · Masa Kageyama (Editor) · 30 Aug 2019

Dear Authors,

All reviews of your manuscript and have been submitted. Please answer them point by point so that I can make my decision.

Best regards

Masa Kageyama

———————————————

---

## Author Comment (AC1) · 13 Sep 2019

Dear Masa, We apologise for the delay in responding to the second review. This was occasioned by the fact that we submitted the paper describing the basic methodology, which we cited as a pre-print (Cleator, S.F., Harrison, S.P., Nichols, N.K., Prentice, I.C., and Roustone, I.: A method for generating coherent spatially explicit maps of seasonal palaeoclimates from site-based reconstructions, arXiv:1902.04973 [math.NA], 2019a), for publication in JAMES. During the review process, a reviewer pointed out that the Gaussian correlation function that we used did not yield a full rank matrix; we therefore moved to using a modified Bessel function that closely matches the behaviour of the original Gaussian function and yields a correlation matrix that is full rank and positive. We have checked that this change does not make a substantial difference to the global

reconstructions presented here and does not change the conclusions of our paper. It has, however, necessitated updating the figures and the text. We needed to do this before responding to the additional reviews. The current version of the revised method paper is available from arXiv (arXiv:1902.04973v2, https://arxiv.org/abs/1902.04973v2). We will upload detailed responses to the two additional reviews shortly.

———————————————————

---

## Author Comment (AC2) · 19 Sep 2019

We thank Michel for his comments on the paper and are glad to have the chance to respond to them.

Mathematical details of the technique applied in this study are available in Cleator et al., 2019a. This is an arXiv preprint. It is not clear whether the latter is intended for a peer-reviewed journal or whether it was part of a thesis examination.

Response: This article is currently in review with the Journal of Advances in Modeling Earth Systems. As a result of the review process, it was pointed out that the Gaussian correlation function we used did not yield a full rank matrix; we have therefore moved to using a modified Bessel function that closely matches the behaviour of the

original Gaussian function and yields a correlation matrix that is full rank and positive. We have checked that this change does not make a substantial difference to the global reconstructions presented here (although it changes some numbers slightly, and we have amended the text to reflect this) and does not change the conclusions of our paper. The revised method paper is available from arXiv (arXiv:1902.04973v2, https://arxiv.org/abs/1902.04973v2) and we hope will soon be available in JAMES. We have updated the figures and the text here to reflect the use of the revised function.

There are a number of points in this approach which deserve discussion. For this reason it would have been better to see these method details in the Climate of the Past paper, so that the paper, the review, and possible responses constitute a self-contained contribution.

Response: We wanted to focus the discussion here on the results (i.e. the reconstructions of LGM climate) rather on the mathematical details of the method. These details should shortly be available in JAMES and are given in the pre-print article. However, we agree that it would be worth expanding the section on the application of the data assimilation method and the choice of length scales (section 2.4) to provide more details. We propose to modify this section as follows:

Variational data assimilation techniques provide a way of combining observations and model outputs to produce climate reconstructions that are not exclusively constrained to one source of information or the other (Nichols, 2010). We use the 3D-variational method, described in Cleator et al. (2019a) to find the maximum a posteriori estimate (or analytical reconstruction) of the palaeoclimate given the site-based reconstructions and the model-based prior. The method constructs a cost function, which describes how well a particular climate matches both the site-based reconstructions and the prior, by assuming the reconstructions and prior have a Gaussian distribution. To avoid sharp changes in time and/or space in the analytical reconstructions, the method assumes that the prior temporal and spatial error correlations are derived from a modified Bessel function, in order to create a climate anomaly field that is smooth both from month to

month and from grid cell to grid cell. The degree of correlation is controlled through two length scales: a spatial length scale that determines how correlated the error in the prior is between different geographical areas, and a temporal length scale that determines how correlated it is through the seasonal cycle. The site-based reconstructions are assumed to have negligible correlations at these space and time scales. The maximum a posteriori estimate is found by using the limited memory Broyden-Fletcher-Goldfarb-Shanno method (Liu and Nocedal 1989) to determine the climate that minimises the cost function. A first order estimate of the analysis error covariance is also computed. An observation operator based on calculations of the direct impact of [CO2] on water-use efficiency (section 2.3) is used in making the analytical reconstructions. The prior is constructed as the average of eight LGM climate simulations (section 2.2). We use an ensemble of different model responses to the same forcing to provide a series of physically consistent possible states, which can be viewed as perturbed responses and provide the variance around the climatology provided by the ensemble average. The prior error correlations are based on a temporal length scale ($L_t$) of 1 month and a spatial length scale ($L_s$) of 400km. Cleator et al., (2019a) have shown that a temporal length scale of 1 month provides an adequately smooth solution for the seasonal cycle, both using single sites and over multiple grid cells, as shown by the sensitivity of the resolution matrix (Menke, 2012; Delahaies et al., 2017) to changes in the temporal length scale. Consideration of the spatial spread of variance in the analytical reconstruction shows that a spatial length scale of 400km also provides a reasonable reflection of the large-scale coherence of regional climate change.

Additional references: Liu, D. C., & Nocedal, J. (1989). On the limited memory BFGS method for large scale optimization. Mathematical Programming, 45 (1), 503–528. doi: 10.1007/BF01589116 Delahaies, S., Roulstone, I., & Nichols, N. (2017). Constraining DALECv2 using multiple data streams and ecological constraints: analysis and application. Geoscientific Model Development (Online), 10 (7). doi: 10.5194/gmd-10-2635-2017 Menke, W. (2012). Geophysical data analysis: Discrete inverse theory (Matlab 3rd ed.). Cambridge, Massachusetts: Academic Press.

Unlike what (roughly) obtains when using time series of a numerical weather prediction system, there is a priori no guarantee that the covariance matrix of a multi-model ensemble produces modes which satisfy "physical consistency". Why would we expect that the inter-model differences provide knowledge about how different variables should co-vary?

Response: Our argument here is that the average response of all the models gives a measure of climatology. Numerical weather prediction uses ensembles of perturbed responses to provide a series of physically-consistent possible states, although there are examples of using multiple models to form an ensemble (see e.g. Johnson and Swinbank, 2009 - https://rmets.onlinelibrary.wiley.com/doi/pdf/10.1002/qj.383). Here we use an ensemble of different model responses to the same forcing, which can be viewed as producing perturbed responses to the general climatology. We have added a sentence in the method text (given above) to make this argument clearer.

In principle, a "prior" encodes what we a priori believe the climate could be. The authors have then chosen to mask regions with little update by observations, and leave visible the grid points where the observations have seriously shifted the prior. This seems at first sight reasonable because the idea is to focus on the pollen reconstructions and not on the PMIP3 output. Yet, at face value, this approach is inconsistent with a Bayesian interpretation. Grid points of strong update are associated, in the Bayesian interpretation, with a very small marginal likelihood (a wrong prior means a wrong model).

Response: We are starting from the assumption that the pollen-based reconstructions are more likely to be correct than the model simulations; but that the model simulations provide us with physically-consistent relationships in space and time (which cannot be obtained from the pollen). This comment is partly due to a misunderstanding about the masking, which is in fact determined by the variance rather than the absolute change. Only areas with an improved variance are shown (i.e. left unmasked). This means that the likelihood that these reconstructions represent the true climate is significantly

improved from the prior. This only happens if the variance in the observations is small and the variance in the prior is big. By using both local and global measures of the variance in the prior, we avoid a situation where the variance in the prior is small but shows a different signal from the pollen-based reconstructions. We will expand the text to make this clearer (lines 277-278) as follows:

The reliability of the analytical reconstructions was assessed by comparing these composite errors with the errors in the prior. We masked out cells where the inclusion of site-based reconstructions does not produce an improvement of > 5% from the prior. Since this assessment is based on a change in the variance, rather than absolute values, this masking removes regions where there are no pollen-based reconstructions or the pollen-based reconstructions have very large uncertainties.

To what extent should we be concerned that the posterior variance remains influenced by the prior variance? Indeed, mathematically, the posterior variance is bounded by the prior variance, which — if we admit the models are really off — is meaningless.

Response: The posterior variance is influenced (though not bounded or limited by) by the prior variance. However, since areas that have a small change to prior covariance are masked out, only areas with pollen-based reconstructions with low variance are used in the reconstruction. Hence, the prior variance only influences the posterior variance in areas that are well constrained by observations. Furthermore, since the prior variance is based partly on the global variance for each variable, the only way to have a large prior variance affecting the posterior variance is for all models to agree well globally and locally and the observation to have a low variance such that the posterior variance has improved upon the prior variance change by over the 5% cutoff. We agree that the choice of the cutoff is somewhat arbitrary (as we state in the discussion section,(lines 406-412), though guided by examination of the impact of this cutoff on the reconstructions, and that it would be useful to develop an objective way of determining an appropriate limit. We will expand the discussion further, to suggest ways forward here (lines 406-412) as follows:

We have used a <5% reduction in the analytical uncertainty compared to prior uncertainty to identify regions where the incorporation of site-based data has a negligible effect on the prior as a way of masking out regions for which the observations have effectively no impact on the analytical reconstructions. The choice of a 5% cut-off is arbitrary, but little would be gained by imposing a more stringent cut-off at the LGM given that many regions are represented by few observations. A more stringent cut-off could be applied for other time intervals with more data. We avoid the use of a criterion based on the analytical reconstruction showing any improvement on the prior because this could be affected by numerical noise in the computation. Alternative criteria for the choice of cut-off could be based on whether the analytical reconstruction had a reduced uncertainty compared to the pollen-based reconstructions or could be derived by a consideration of the condition number used to select appropriate length scales.

To what extent the prior covariance (link between different variables) may still be trusted at all if the models are so wrong? This remark strengthens the original concern about the physical meaning of the covariance matrix, even when the prior is only mildly updated. What is the advantage of this approach over a mere Gaussian interpolation (flat climate prior), which in this case might turn out to be more reliable and free of the dubious claim of "physical consistency"?

Response: We acknowledge that the climate models may not be correct, for example because the LGM simulations do not include all of the necessary forcings or show weak responses to these forcings. However, analyses of the PMIP simulations indicate that while the models show differences of both magnitude and sign in some regions, the overall LGM to present change is broadly consistent with what we know from observations. It is worth pointing out that many of these regional problems are associated with model dynamics rather than thermodynamics, which suggests that the models can be used to ensure physical consistency between surface variables. We try to overcome the problem of "all models being consistent but wrong" at a regional scale by combining global and local uncertainties to produce the uncertainty on the prior. In revising

the section describing the variational approach (see above), we have tried to make our logic clearer here.

Were the length scales tested by some form of cross-validation (e.g. leave-one-out), or were they merely chosen because they are a priori reasonable?

Response: We did not use cross-validation to evaluate the choice of length scales, but instead we based the choice of length scales on sensitivity experiments (as described in the arXiv pre-print). Effectively we ran a series of tests to see how different choices affect the resolution matrices and the condition number. We selected a spatial length scale that provided a reasonable reflection of the large-scale coherence of regional climate change and also ensured that the covariance matrix was well-conditioned for inversion, and a temporal length scale that limited overlap between successive months. The selected length scales seem reasonable; for example, the spatial scale corresponds to a situation where there is little overlap between data points assuming an average catchment size for the pollen records on which the original reconstructions were based. Similarly, the selected temporal length scale produces plausible-looking seasonal cycles of temperature. We have expanded the text describing the application of the variational method (see above) to clarify how the length scales were chosen based on these sensitivity tests and a post-hoc evaluation of plausibility.

The arXiv paper provides the definition of the moisture index. It should be repeated here (moisture index is currently introduced l. 297 without definition)

Response: We apologise for not defining MI at first use; it is in fact currently defined at line 155. In the present context the reference to MI is inappropriate because the text refers to a generic control by moisture availability rather than a specific index. We also note there was a crucial comma missing in this sentence! We will alter the text here to read: which is generally taken into account by process-based ecosystem models, but not by statistical models, using projected changes in vapour pressure deficit or some measure of plant-available water

The authors should consider providing a link to supporting code. The maps are currently provided as University of Reading dataset (with a doi) but its lifecycle is detached from the present contribution. A dataset consistent with the current Climate of the Past submission, reflecting a possible response to concerns of the reviewers, might best be included as supplementary information. Have we lodged code somewhere?

Response: The data used to generate the maps are lodged at the University of Reading repository, with a DOI. This allows external users to generate their own maps and their own analyses using the reconstructions. A revised version of these data, reflecting minor changes in the data as a consequence of using a Bessel function, has now been lodged at the repository. The two data sets are linked, so that external users are directed to the updated version of the data set. We do not envisage any changes to the data set as a result of review of this CoP submission, but if there are further changes to the data set then the current data set can be updated and again linked. Thus, the data provided in the repository are indeed constantly linked to the lifecycle of the product. The code used to generate the reconstructions has been lodged at Zenodo, and we will provide the DOI for this code in the revised ms. We will add a data availability section to the ms as follows: Data availability: the gridded data for the LGM reconstructions are available from http://dx.doi.org/10.17864/1947.206; the code used to generate these reconstructions is available from (10.5281/zenodo.3445166).

It is important to distinguish the notion of variance from the notion of uncertainty. They are not synonymous. Variance describes the second momentum of a distribution; uncertainty is a reference to an identified lack of knowledge. Only when the distribution is assumed reflects our knowledge of a given quantity is it legitimate to identify both.

Response: We agree that the use of terminology here is inaccurate and we need to be more precise. However, uncertainty is not simply an identified lack of knowledge! It is also used to refer to the limits on the precision of knowledge (as in the case where we talk about the uncertainties attached to a pollen-based climate reconstruction, which are partly a function of our ability to define precise relationships with existing training

data sets). We have corrected the ms throughout to ensure that we use variance and uncertainty appropriately. We have made the following specific changes: l.34 error changed to uncertainty l.134 error changed to uncertainty l.268 error changed to variances l.269 error changes to covariances l.272-278 error changed to variance l.282 error changed to variances l.283 error changed to variances (We have also changed this for Figure 3 in the caption list section) l.147 uncertainty change to variances Figure 2 caption, uncertainty changed to variances Figure 3 caption, uncertainty changed to variances l.406 uncertainty changed to variance

Multi-model ensembles, in general, cannot be said to capture our knowledge of the state of climate at a given time. For this reason, I would argue not to call the PMIP3 covariance a "background uncertainty".

Response: We agree that models are not the only source of information about the state of the climate at a given time, and indeed our approach makes the assumption that the pollen-based reconstructions are more likely to represent the true state of the climate. We agree that the models may be wrong because they do not include all the appropriate forcings, because the response to these forcings is too weak, or because of inappropriate treatment of key feedbacks. We also agree that not all models are equally good (or bad) and that in an ideal world a prior should be reconstructed based only on an ensemble of well-validated models. However, the point of using climate models here is to provide a way of deriving physically consistent relationships between climate variables, given that we do not have reconstructions of all of the seasonal variables everywhere. Furthermore, there are comparatively few LGM simulations available and using a more limited number of "more likely to be correct" simulations to create the prior (and estimate its variance) does not seem to be a good option. In the future, it might be possible to combine PMIP3 and PMIP4 simulations to create a more robust/plausible prior, but this is currently not possible.

The legend of Figure 2 clearly identifies "uncertainties" with "standard deviation of the non-dimensionalised multi-model ensemble" but this seems inadequate to me. Adding

to the confusion, different qualifiers occur throughout the text: "explicit uncertainty" (l. 97), "analytical uncertainty" (l. 406), and, on Figure 3, "grid-based errors in the prior" and "global uncertainty".

Response: We agree that we have not been consistent about the terminology, and particularly the use of terms such as uncertainty, error and variance. We will revise the manuscript so that we are consistent about the terminology, and we will make sure that our use of terms such as explicit uncertainty and analytical uncertainty is clearer for the reader. The changes made are listed in response to the earlier comment about the confusion between uncertainty and variance.

As the uncertainty quantification seems to be a selling point of the present article, the assessment should be more open and transparent about sources of uncertainty, and discuss which of theses sources can be quantified and how. For example, little is said about uncertainties introduced by the CO2 physiological correction. Is it guaranteed to be accurate?

response: There are three basic sources of uncertainty: the pollen-based reconstructions, the construction of the prior, and the uncertainties associated with our implementation of the method. While we have addressed the uncertainties associated with the first two, we agree that the methodological uncertainties were not as well addressed in this paper (although they are discussed in the Prentice et al., 2017 paper from which we derived the CO2 correction approach, and in the arXiv pre-print). The expanded description of the variational method (see above) is now more explicit about potential uncertainties associated e.g. with choice of length scales and cut-offs. For the CO2 correction, we made a series of sensitivity analyses in the Prentice et al. (2017) paper to determine the impact of uncertainties (or errors) in the input parameters. These sensitivity tests showed that the magnitude of the correction was insensitive to the reconstructed temperature, the reconstructed change in temperature relative to the modern reference, or the reconstructed moisture level. The magnitude of the correction is highly sensitive to the level of CO2 specified, but this is well-constrained from the icecore records. We will expand the text in the discussion of the CO2 effect to make this clearer (lines 378-385), as follows:

…. differences in water use efficiency of different PFTs can be almost entirely accounted for by a single equation, as proposed here. Sensitivity analyses show that the numerical value of the corrected moisture variables (MI, MAP) is dependent on the reconstructed values of these variables, but is insensitive to uncertainties in the temperature and moisture inputs (Prentice et al., 2017). The strength of the correction is primarily sensitive to [CO2], but the LGM [CO2] value is well constrained from ice-core records. The response of plants to changes in [CO2] is non-linear (Harrison and Bartlein, 2012), and the effect of the change between recent and pre-industrial or mid-Holocene conditions is less than that between pre-industrial and glacial conditions. Nevertheless, it would be worth taking the [CO2] effect on water-use efficiency into account in making reconstructions of interglacial time periods as well.

The strategy for identifying grid points with little posterior update explained l. 406 is not quite clear. Why not consider a Kullback-Leibler divergence? At the risk of repeating myself, I am concerned about the (meaningless) residual influence of the prior variance and covariance in cases where the prior is effectively discarded by the observations.

Response: As we have explained above in response to the question about masking (and will clarify in the text, lines 277-278), for each variable in each grid cell, we calculate the percentage change of variance between the prior and posterior. We then mask away variables where there is a less than 5% increase in variance. We do not use the Kullback-Leibler divergence approach because this requires the calculation of covariance. However, the two approaches will likely not yield results that are very dissimilar.

This is a minor comment, but the comparison with Goosse et al. 2006 is perhaps slightly misleading. The Goosse et al. purpose was dynamic reconstruction, while the purpose of the present contribution is to provide a diagnostic reconstruction. In

passing, Goosse (2006) did not use a "Kalman particle filter" (whatever it means). Goosse et al. used what they called an "optimal realisation" iteration, which can be interpretated as a highly degenerate form of particle filter. Dubinkina et al. 2011, doi 10.1142/S0218127411030763, adopted a more standard particle filter.

Response: The reference to the Kalman filter is somewhat misleading, although as you point out the approach used by Goosse et al. (2006) can be considered a form of particle filter. Our point here is that filters that select from model output are inherently constrained by the model output, whereas variational approaches can go beyond the values produced by the model. We have changed the wording of the text (line 420-422) to make our main point clearer:

Particle filter approaches (e.g. Goosse et al., 2006; Dubinkina et al., 2011) produce dynamic estimates of palaeoclimate, but particle filters cannot produce estimates of climate outside the realm of the model simulations.

This said, the argument that the variational approach produces maps outside the realm of climate simulations is a double-edged sword. The variational approach assumes Gaussian distributions, and is mathematically equivalent to a Laplace approximation of arbitrary distributions. This is this approximation which allows generating posterior distributions far from the prior. But, in this case, sound Bayesian interpretation should lead us to treat such posterior as utterly suspicious.

Response: It is not clear why a posterior distribution that is far from the model-based prior is utterly suspicious, if being far from the prior reflects the fact that the observational constraints are strong. We are not pretending that there should be equal weight given to the model-based prior and the pollen-based reconstructions, only combining the two and drawing on their individual strengths produces a more reliable estimate of the "true" climate state. Our approach is specifically designed to permit analytical reconstructions that are far from the model-based prior, if this is consistent with the observations and those observations have low variance.

line 384 : It is said that it "would be worth taking [changes in length scales] into account." I would advise either deleting this sentence, or giving more substance to the claim. For example, have you already performed some sensitivity experiments.

Response: The cited text is not talking about changes in length scales, but rather about the necessity to take the CO2 correction into account in making reconstructions of interglacial climates. We have amended this sentence to make this clear, as follows:

Nevertheless, it would be worth taking the [CO2] effect on water-use efficiency into account in making reconstructions of interglacial time periods as well.

Is the very first paragraph really necessary?

Response: Strictly speaking, it should not be necessary, especially for a palaeoclimate audience. However, this does seem to be a point which is largely ignored by many climate modelling centres worldwide, and at least one of the authors (SPH) thinks it bears repeating. We can remove the paragraph if the editor disagrees.

There is room for improving wording accuracy. In what sense is the benchmark "robust" (l. 37) ? l. 97: You write: "explicit uncertainties attached to it". Did you mean "uncertainties explicitly attached" ? Avoid, where possible, the phrase "in terms of" or "means that" (ll. 321 - 326, in particular, need rewording). What is meant by a "statistical reconstruction method" l. 370 (the present exercise is a statistical reconstruction isnt'it ?).

Response: We have been through the manuscript and tightened up the wording. With respect to the specific sentences above, we have made the following changes:

L 37: Thus, the new reconstructions provides a benchmark created using clear and defined mathematical procedures that can be used for evaluation of the PMIP4/CMIP6 entry-card LGM simulations.

L. 97: However, there has so far been no attempt to produce a physically consistent, multi-variable reconstruction which provides the associated uncertainties explicitly.

L 321 et seq.: There are systematic differences, however, between the analytical reconstructions and the pollen-based reconstructions of moisture-related variables (MAP, MI) because the analytical reconstructions take account of the direct influence of [CO2] on plant growth. The physiological impact of [CO2] leads to analytical reconstructions indicating wetter than present conditions in many regions (Figure 5a, Figure 5b), for example in southern Africa where several of the original pollen-based reconstructions show no change in MAP or MI compared to present, but the analytical reconstruction shows wetter conditions than present. In some regions, incorporating the impact of [CO2] reverses the sign of the reconstructed changes. Part of northern Eurasia is reconstructed as being wetter than present, despite pollen-based reconstructions indicating conditions drier than present (both in terms of MAP and MI), as shown by SI Figure SI 3. The relative changes in MAP and MI are similar across all sites (Figure 5c), implying that the analytically reconstructed changes are driven by changes in precipitation rather than temperature.

L 370: Statistical reconstruction methods that use modern relationships between pollen assemblages and climate under modern conditions (i.e. modern analogues, transfer functions, response surfaces: see Bartlein et al., 2011 for discussion) cannot account for such effects.

Figure 5: Shouldn't "pre-industrial reference" be preferred over the vague wording "original" as x-axis label?

Response: We agree that the axis labels on this Figure are not clear. These plots contrast the original pollen-based reconstructions of MI and MAP with analytical reconstructions before (circles) and after (crosses) the CO2 effect is taken into account. We will change the axis labels to read: Pollen-based MI and Pollen-based MAP. We will also expand the caption to make this clearer, as follows:

Figure 5: Impact of CO2 on reconstructions of moisture-related variables. The individual plots show (a) the change in moisture index (MI) and (b) the change in mean

annual precipitation (MAP) compared to the original pollen-based reconstructions for the LGM when the physiological impacts of [CO2] on water-use efficiency are taken into account. The third plot (c) shows the relative difference in MI and MAP as a result of [CO2], shown as the percentage difference between the no-[CO2] and [CO2] calculations.
* * *

---

## Author Comment (AC3) · 19 Sep 2019

Response to Anonymous Referee #3 Review

The paper is quite short and lacks any detailed evaluation of the resultant product. The community's use of this new data product would in my opinion be aided by a more in-depth evaluation of the properties of the reconstruction. It's not clear how important the choices around the assimilation formulation are for the final reconstruction. Specifically the section around lines 268-278 should in my opinion be spelled out and the sensitivity to these choices evaluated.

Response: It is unclear what kind of evaluation of the product the reviewer envisages, given that there is no global ground-truth data set other than the pollen-based reconstructions themselves. We have already pointed out (lines 317-321) that the analytical reconstructions of temperature are close to the Bartlein et al. (2011) data set, both in terms of magnitudes and spatial patterns. The differences between the analytical reconstructions and the Bartlein et al. (2011) reconstructions of moisture variables are a consequence of the fact that statistical techniques based on modern pollen-climate relationships cannot account for $CO_2$-induced changes in water-use efficiency. In terms of the impact of methodological choices, the major issue here is the choice of length scales. We have made sensitivity analyses to examine the implications of the choice of length scales, and this was discussed in the arXiv preprint. In expanding the description of the application of variational techniques here (see text in response to Michel Crucifix's review) we have commented further on this.

The statistical methodology that forms the basis of this study is also not described here but in a arXiv article. I'd like to see more of this brought into the present manuscript to make it self-contained.

Response: This is a point raised by Michel Crucifix in his review. We have now modified the text describing the application of the variational method to include a fuller description of our approach. As we point out in the response to Michel Crucifix's review, the full details of the method are now in review for JAMES and we have made the post-review version of this paper available on arXiv.

Line 127: define MI here.

Response: The reference to MI is inappropriate in the present context because the text refers to a generic control by moisture availability rather than a specific index. We also note there was a crucial comma missing in this sentence! In response to Michel Crucifix's review, we have modified this text to read:

which is generally taken into account by process-based ecosystem models, but not by statistical models, using projected changes in vapour pressure deficit or some measure of plant-available water

[Figure]

Line 209: modelsfor -> models for

Response: We have corrected this typo.

Line 252-253: I think it might be appropriate to bring some/all of this methodology into the present text, as discussed above. Response: We have modified the text here to provide more detail about the method. Please see proposed revised text given in the response to Michel Crucifix's review.

One question that arises from briefly reading the methodology paper, relates to figure 1 in the arXiv article. Here the assimilation appears not satisfy the pollen-inferred MTCO. Is this because the prior (from the models) is relatively consistent, and so doesn't allow the assimilation to get that cold? Does this happen when applied to the pollen data here?

Response: Figure 1 in the arXiv pre-print does not show a real situation but was designed (as explained in that paper) to illustrate the procedure. In general, the pollen-based reconstructions of MTCO are further away from the model-based prior then summer temperature measures. If the variance in the pollen-based MTCO reconstructions is small, then the analytical reconstructions will be close to the pollen-inferred MTCO. If there is high uncertainty in the pollen-based reconstructions, then the analytical reconstructions are not strongly constrained by these reconstructions and will be further away. This makes intuitive sense because we do not want to rely on pollen-based reconstructions if there is large uncertainty. Thus, it is possible for the assimilation to produce cold results but only if there is tight agreement between the observations about the magnitude of the cooling.

How do we interpret these choices, given that the climate models themselves could feasibly be systematically biased, e.g. through not including aerosols, or using modern vegetation distributions? How have you addressed the possible systematic bias in the models and hence in your prior?

Response: It is possible that the models show a systematic bias because they do not include all of the appropriate forcings for the LGM climate. We assume that such a systematic bias would primarily influence the magnitude of changes rather than the physical relationship between variables or across space. The presence of a systematic bias is therefore not important because the pollen-based reconstructions effectively correct for any systematic biases in the model-based prior, providing the pollen-based reconstructions have low uncertainty. One of the reasons that we discuss in the paper for adopting a variational technique, rather than some kind of filtering, is that this approach means that the analytical reconstructions can go beyond the range of the simulated climate.

Line 268-276: This section seems crucial to me, but is not clearly described. Please include the mathematical formulation used and a justification for choices made.

Response: We have expanded the text describing the application of the variational method, including a description of the composite errors. Please see revised text included in the response to Michel Crucifix's review.

Lines 276-278: Do you mean that if the data is too uncertain you mask it based on a 5% criteria? Please could you re-phrase to clarify.

Response: When the change in the variance between the analytical reconstruction and the prior is less than 5%, it does indeed mean that the climate is not well constrained by observations (i.e. that there is high uncertainty in the observations). We have modified this text (and the discussion of the choice of cutoff in the Discussion) to clarify this point. Please see revised text in response to Michel Crucifix's review.

Lines 288: How does your product compare with the original Bartlein et al 2011, and the GCM-based prior? Could you show this?

Response: The GCM-based prior is shown in SI Figure 1 and the original pollen-based reconstructions (from Bartlein et al, 2011 and from Prentice et al., 2017) are shown in

[Figure]

SI Figure 3. Comparison of these figures with the analytical reconstructions shown in the paper in Figure 4 shows the difference with our product. We could add a new set of figures to the Supplementary showing difference maps, if necessary.

How well is the seasonality captured and how does it differ from the simulated seasonality in the GCM prior?

Response: It is not clear what the reviewer is asking here. We have no independent measure of seasonality that can be used to assess the analytical reconstructions. The analytical reconstructions of MTCO and MTWA, the difference between which is the measure of the strength of temperature seasonality, are only shown when the pollen-based reconstructions contain sufficient information to modify the model-based prior and thus when the uncertainty in the pollen-based reconstructions is small. We could produce maps showing the temperature seasonality from the analytical reconstructions and the model ensemble (and indeed the difference between them) but it is not clear what these would add to the manuscript.

---

## Author Comment (AC4) · 19 Sep 2019

Response to review 1

We thank the reviewer for their positive and helpful comments on our manuscript.

Comment: This is a relatively short paper that relies heavily on previously published work for data and methodology, which makes it hard to review in a very detailed fashion, because there is not much data included. The results presented are simply a number of global maps, with very little data in the text or supplementary information. As the previous studies have passed peer review this does not bother me too much, and the paper is clearly and well written. Hence I am happy to recommend publication with minor technical corrections. Response: As the reviewer points out, the pollen-based recon-

structions and the climate model simulations underpinning our reconstruction are in the public domain, and the data assimilation methodology is described in detail in another publication currently in review for JAMES and available at arXiv (arXiv:1902.04973v2, https://arxiv.org/abs/1902.04973v2). The general approach used for the $CO_2$ corrections, which the reviewer describes as a significant contribution, was published in Prentice et al. (2017) – although we provide the equations for the implementation of this approach in the current paper in Appendix 1. Therefore, the new results in this paper are indeed the global maps of reconstructed climate variables. These data are archived and will be made publicly available – however, we realise that it may not have been obvious that the citation to Cleator et al. (2019b) represented the reconstruction data set. We propose to modify the last sentence of the abstract to make it clear that the reconstruction data are available as follows: The new reconstructions will provide a robust benchmark for evaluation of the PMIP4/CMIP6 entry-card LGM simulations and are available at DOI:10.17864/1947.206 We also note that, in response to other review comments, we have now also posted the code used to make the reconstructions at Zenodo: 10.5281/zenodo.3445166 We did not include a Data Availability section in the Discussion paper and we will also rectify this: Data availability: the gridded data for the LGM reconstructions are available from http://dx.doi.org/10.17864/1947.206; the code used to generate these reconstructions is available from (10.5281/zenodo.3445166).

Comment: The definition for the LGM given here is 21±1 ka, and this appears to be because previous work has used this temporal extent. However it is different from, for example, to the range used by Annan and Hargreaves (cited in the paper) of 21±2ka, and recent work on sea level (Ishiwa et al. 2019) suggest the 'real' LGM was 19.1-19.7 ka, with a plateau prior from 20.4-25.9ka, both pushing out past the time interval use in this study. I don't think there is anything in particular to be done about this – just to think about. . . Response: The reviewer indicates that the definition of the LGM used in our paper (21±1 ka) differs from the interval used by Annan and Hargreaves of 21±2ka, and there is recent work on sea level (Ishiwa et al. 2019) which suggests the 'real' LGM was 19.1-19.7 ka, with a plateau prior to this from 20.4-25.9ka. Our

choice of this time interval reflects the fact that the LGM is conventionally defined in PMIP at 21 ka and most of the pollen- based reconstructions of this interval included in the Bartlein et al data set are from the 21±1 ka. We are aware that there is still controversy over the timing of the LGM, with both younger and older ages mooted for the actual maximum ice volume/sea- level lowering (see e.g. Peltier and Fairbanks, 2006; Clark et al., 2009; Lambeck et al., 2014). Even the recent work by Ishiwa et al. (2019) points out that the sea level drop after 19.7 ka was only 10m and that there was a long plateau with stable low sea level prior to this and encompassing the 21 ka interval. Since our aim is to produce a data set for benchmarking new PMIP LGM simulations, which will be run with boundary conditions for 21 ka (Kageyama et al., 2017), the exact date of the LGM is therefore not an issue. However, we agree that there is a difference between the true definition of the LGM and the convention used for modelling purposes, and that this is not clear from our introductory text, so we propose to expand our definition (lines 57-61) as follows: At the Last Glacial Maximum (LGM, conventionally defined for modelling purposes as 21 000 years ago), insolation was quite similar to the present, but global ice volume was at a maximum, eustatic sea level was close to a minimum, long-lived greenhouse gas concentrations were lower and atmospheric aerosol loadings higher than today, and land surface characteristics (including vegetation distribution) were also substantially different from today.

Comment: a general point I guess, that the paper refers to several new studies since the Bartlein paper on which the analysis is based, and there are more. It would be nice to think these could be assimilated into a future dataset to maybe close some of the large 'no data' holes in the results. . . Response: The reviewer points out that we refer to several new studies since the Bartlein paper on which the analysis is based, and there are more, and that it would be nice to think these could be assimilated into a future dataset to maybe close some of the large 'no data' holes in the results. We thoroughly agree that it would be good to plug the gaps, and this will be an effort for the future. The three papers that we cite at lines 361-363 (Flantua et al., 2015; Herbert and Harrison, 2016; Harrison et al., 2016) demonstrate that there are pollen records

available that would plug the gaps, but alas do not pro- vide quantitative reconstructions at these sites. The Izumi and Bartlein, 2016 paper provides an inversion-based reconstruction for North American – this region is already relatively well covered in the Bartlein et al data set. Similarly Mauri et al., 2015 provide a new gridded reconstruction for Europe – again a region that is well covered in the Bartlein et al data set. However, we are aware of new pollen-based quantitative recon- structions embracing the LGM for individual sites (e.g. in Africa, China, Russia, south- ern Europe) and compiling these reconstructions would certainly be a worthwhile effort in the future. Our method also lends itself to combining pollen-based reconstructions with other quantitative estimates of terrestrial palaeoclimate, and again this should be something that is done in the future. We will expand the paragraph describing future possibilities to expand the current data set to spell out some of these opportunities more clearly, as follows:

Some areas are still poorly covered by quantitative pollen-based reconstructions of LGM climate, most notably South America. More pollen-based climate reconstructions would provide one solution to this problem – and there are many pollen records that could be used for this purpose (Flantua et al., 2015; Herbert and Harrison, 2016; Harrison et al., 2016). There are also quantitative reconstructions of climate available from individual sites (e.g. Lebamba et al., 2012; Wang et al., 2014; Loomis et al., 2017; Camuera et al., 2019) that should be incorporated into future data syntheses. It would also be possible to incorporate other sources of quantitative information, such as chironomid-based reconstructions (e.g. Chang et al., 2015) within the variational data assimilation framework.

Additional references Camuera, J., JimeÌĄznez-Moreno, G., Ramos-RomaÌĄn, M.J., GarciÌĄa- Alix, A., Toney, J.L., Anderson, R.S., JimeÌĄnez-Espejo, F., Bright, J., Webster, C., Yanes, Y., JoseÌĄ S. CarrioÌĄn, J.S., 2019. Vegetation and climate changes during the last two glacial-interglacial cycles in the western Mediterranean: A new long pollen record from Padul (southern Iberian Peninsula), Quaternary Science Reviews, 205, 86-105, https://doi.org/10.1016/j.quascirev.2018.12.013. Chang, J.C., Shulmeis-

ter, J., Wood- ward, C., Steinberger, L., Tibby, J., Cameron Barr, C., 2015. A chironomid-inferred summer temperature reconstruction from subtropical Australia during the last glacial maximum (LGM) and the last deglaciation, Quaternary Science Reviews, 122, 282- 292, https://doi.org/10.1016/j.quascirev.2015.06.006. Lebamba, J., Vincens, A., and Maley, J.: Pollen, vegetation change and climate at Lake Barombi Mbo (Cameroon) during the last ca. 33 000 cal yr BP: a numerical approach, Clim. Past, 8, 59-78, https://doi.org/10.5194/cp-8-59-2012, 2012. Loomis, S. E., Russell, J. M., Verschuren, D., Morrill, C., De Cort, G., Sinninghe Damstel̨, J. S., . . . Kelly, M. A. (2017). The trop- ical lapse rate steepened during the Last Glacial Maximum. Science advances, 3(1), e1600815. doi:10.1126/sciadv.1600815 Wang, Y., Herzschuh, U., Shumilovskikh, L. S., Mischke, S., Birks, H. J. B., Wischnewski, J., Bol̀hner, J., Schlul̀tz, F., Lehmkuhl, F., Diekmann, B., Wul̀nnemann, B., and Zhang, C.: Quantitative reconstruction of precipi- tation changes on the NE Tibetan Plateau since the Last Glacial Maximum – extending the concept of pollen source area to pollen-based climate reconstructions from large lakes, Clim. Past, 10, 21-39, https://doi.org/10.5194/cp-10-21-2014, 2014.

Comment: L59: change to 'lower, atmospheric aerosol. . .' Response: We will make this change.

Comment L321: comma after 'however' (I think?) Response: We will make this change.